Conducting perception research over the internet: a tutorial review

Woods Andy T. 1 2 andytwoods@gmail.com
Velasco Carlos 1
Levitan Carmel A. 3
Wan Xiaoang 4
Spence Charles 1
1 Crossmodal Research Laboratory, Department of Experimental Psychology, University of Oxford , United Kingdom
2 Xperiment , Surrey , United Kingdom
3 Department of Cognitive Science, Occidental College , Los Angeles , USA
4 Tsinghua University , Beijing , China
Thompson Steven
Electronic publication date: 2015 Jul 23
Publication date: 2015
Volume: 3
Electronic Location ID: e1058
Received 2015 Mar 26; Accepted 2015 Jun 7
Copyright: © 2015 Woods et al.
Copyright year: 2015
Copyright holder: Woods et al.
License: This is an open access article distributed under the terms of the Creative Commons Attribution License, which permits unrestricted use, distribution, reproduction and adaptation in any medium and for any purpose provided that it is properly attributed. For attribution, the original author(s), title, publication source (PeerJ) and either DOI or URL of the article must be cited.
License URL: https://creativecommons.org/licenses/by/4.0/

Keywords: Perception, Citizen science, Prolific academic, Mechanical Turk, Internet-based testing, Haxe

Funding: AHRC AH/L007053/1 CS received funding from the AHRC for the Rethinking the Senses project (AH/L007053/1). The funders had no role in study design, data collection and analysis, decision to publish, or preparation of the manuscript.

==============================
This article provides an overview of the recent literature on the use of internet-based testing to address important questions in perception research. Our goal is to provide a starting point for the perception researcher who is keen on assessing this tool for their own research goals. Internet-based testing has several advantages over in-lab research, including the ability to reach a relatively broad set of participants and to quickly and inexpensively collect large amounts of empirical data, via services such as Amazon’s Mechanical Turk or Prolific Academic. In many cases, the quality of online data appears to match that collected in lab research. Generally-speaking, online participants tend to be more representative of the population at large than those recruited for lab based research. There are, though, some important caveats, when it comes to collecting data online. It is obviously much more difficult to control the exact parameters of stimulus presentation (such as display characteristics) with online research. There are also some thorny ethical elements that need to be considered by experimenters. Strengths and weaknesses of the online approach, relative to others, are highlighted, and recommendations made for those researchers who might be thinking about conducting their own studies using this increasingly-popular approach to research in the psychological sciences.

Introduction

Over the last few years, the rapid growth of online research has revolutionized the way in which many experimental psychologists choose to conduct (at least some of) their research. On the one hand, it holds the promise of allowing the researcher to go well beyond the typical constraints of the Western, Educated, Industrialised, Rich, and Democratic (WEIRD, see Henrich, Heine & Norenzayan, 2010) pools of participants who form the basis for the vast majority of psychological research. Internet-based testing also opens up the possibility of conducting research cross-culturally (e.g., Knöferle et al., 2015; Woods et al., 2013). Furthermore, the experience of many of those researchers who have started to work/publish in this area is that relatively large numbers of participants (>100) can be collected in a comparatively short space of time (e.g., in less than 24 h, and often in less than 1 h) at relatively low cost (1-2 USD/participant/10 min). Generally speaking, such data collection can be achieved with relatively little effort on the part of the experimenters concerned.

On the downside, however, concerns have been expressed about the lack of control over certain factors, such as the inevitable lack of control over the precise parameters of stimulus presentation (for example, screen resolution/display characteristics), not to mention the lack of experimenter supervision of the participants while taking part in these studies. Another issue of concern is just how often supposedly anonymised data makes its way onto the web, whilst still containing details that can indirectly, and often directly, reveal the identity of the participant.

Nevertheless, despite these various concerns and limitations, there has been a rapid and dramatic growth in the number of studies that have been published using online testing over the last few years (see Fig. 1). We argue that the far larger sample sizes that one typically attracts when engaged in online (as opposed to lab-based) testing, and the much broader diversity of such samples, can more than make up for many of the lacks of control that one is faced with as an experimenter. Indeed, conducting large-scale studies online, in combination with lab-based experiments offering finer control over the testing situation and stimuli, may be an attractive, not to mention economic strategy for a variety—or indeed perhaps the majority—of future psychological research in the area of perception. For the time being, though, such studies are limited to the delivery of visual and auditory stimuli.

Figure 1 Articles per year found on the Web of Science prior to 2015.

The number of articles found on the Web of Science prior to 2015 with the search term ‘Mechanical Turk’ within the ‘psychology’ research area (search conducted on 12th March, 2015).

Target audience, and outline of the present article

In this article, our goal is to provide an up-to-date overview of the pros and cons associated with conducting research online for those perception scientists who are considering supplementing their own lab-based research with this powerful new research methodology (for a general overview of online research for psychology researchers, see Gosling & Mason, 2015). We have focused on present day issues that have mostly arisen since 2011. We flesh out some well-discussed issues in this article ourselves to provide a broad base for the interested reader and critically evaluate the challenges and benefits of online research. First, we highlight how much more representative of the population at large online participants are as compared to their lab-based counterparts, as well as how rapid and economical the collection of data can be online. Following on from this, we explore the various concerns that have been raised with regard to online research, focusing on timing-related issues and how the wide variety of hardware/software that may be used by one’s participants can give rise to data problems. The common concern about the lack of supervision of the participants themselves is also dealt with. Although warranting a paper unto itself, we briefly touch on some of the ethical issues pertaining to online research. We also provide an overview of the main online testing platforms that are currently available to researchers. Finally, we end by drawing some general conclusions and highlighting what we see as the most promising opportunities for future research.

Benefits of Conducting Research Online

Online research (conducted on a computer with access to the internet) has a number of potential benefits over more traditional lab-based studies, which will be evaluated in this section (our focus is primarily computer-based research, but we briefly mention smartphones in a later section). In particular, we discuss how online research can profit from more representative and diverse samples of participants, as well as the more efficient collection of large amounts of data, and simpler participant payments.

Easy access to large pools of participants

One of the most important advantages associated with conducting online research is the speed and ease with which large amounts of data can be collected. In lab-based experiments, researchers typically test participants individually or in small groups over a period of several days, weeks, or even months. Unfortunately, this in-person testing of participants can introduce noise, attributable to, for instance, differences in the way in which the task is explained (though see the article by Mirams et al., 2013, where the researchers attempted to avoid this issue by, amongst other things, making sure that each participant received their instruction by means of an audio-recording) or even basic demographic differences can influence performance on psychological tasks (e.g., Marx & Goff, 2005; Rumenik, Capasso & Hendrick, 1977). Perhaps most pertinently, research assistants/researchers can provide subtle unintentional cues to the participants regarding how to respond to the task at hand (e.g., see Doyen et al., 2012; Intons-Peterson, 1983; Orne, 1962). As Orne noted a little over half a century ago, there is a social psychological element to any in-lab psychology study (perhaps surprisingly, this has not received much attention in recent years). Furthermore, the scheduling of participants takes time, and depending on the specific participant pool, there may be a significant number of participants who do not turn up or else who turn up late to their appointed experimental session. That said, paid for tools such as SonaSystems and Experimetrix nowadays help by automating much of the sign-up process and can also send out reminder emails (https://www.sona-systems.com/, http://www.experimetrix.com/; see also the soon to be released open-source LabMan toolbox, https://github.com/TheHandLaboratory/LabMan/). Another drawback of much of the lab-based research is that it can be difficult to run multiple participants in parallel, because of experimenter constraints, as well as limits on experimental set-ups/space.

By contrast, with online research, when utilizing the appropriate recruitment platform (the focus of the next section), massive numbers of people can undertake a study at any time. What’s more, the availability of participants is not limited by the vagaries of the academic year, with participation in many university settings being much more prevalent in term-time than out of term-time (unfortunately compounding this issue, students who receive course credit as opposed to payment for taking part in studies are both less motivated and have been shown to display less sustained attention at the end of the term as compared to the start, Nicholls et al., 2014). Note that outside of term-time there are more individuals participating in studies online, which in all likelihood correlates with an increased number of students looking to earn some money (R LaPlante, pers. comm., 2015). There can also be severe challenges associated with scaling up one’s sample sizes in the lab setting, whereas online, the pool of potential participants would appear to be more than large enough to adequately address most questions (Mason & Suri, 2012). Another practical benefit of conducting research online is that the payment of participants can often be automated; that is, the researcher need only make one payment instead of many, and does not need to collect hundreds of individual receipts from participants, minimising their interaction with their financial department.

Recruitment platforms

There are several online resources for the recruitment of participants online, with perhaps the most well-known being Mechanical Turk (MTurk; we detail some of the characteristics of platforms popularly used for behavioural research in Table 1; for an overview of this platform from the New Yorker, do watch this recent video: http://www.newyorker.com/culture/culture-desk/video-turking-for-respect). Although this platform is primarily aimed at letting those working in industry recruit many individuals to do tasks related to business such as categorising photos or rating website content (see also CloudCrowd, Innocentive, MicroWorkers and oDesk; Chandler, Paolacci & Mueller, 2013; as well as Peer et al., 2015, on how good some of these sites are when it comes to conducting behavioural research), the last few years have seen an increasing number of psychological studies starting to use the service (e.g., Crump, McDonnel & Gureckis, 2013; perhaps <5% of MTurk usage is academic-related, Sheehan, 2015). In 2014, Mechanical Turk claimed to have half a million individuals registered (Paolacci & Chandler, 2014; http://demographics.mturk-tracker.com provides a live feed of several MTurk demographic characteristics). However, more recent research suggests that the active potential (US) participants available for a typical study are more likely to number only ten thousand (N Stewart, C Ungemach, AJL Harris, DM Bartels, BR Newell, pers. comm., 2015).

Unfortunately, in the summer of 2014, Mechanical Turk stopped allowing new ‘requesters’ (individuals wanting others to complete a task), and new ‘workers’ in 2012 (participants) to sign-up who did not have sufficient credentials identifying them as residing in the United States,1 such as US bank accounts and Social Security Numbers (do see http://ai.reddit.com/r/mturk/#ai, for some personal testimonials on the issue). Consequently, many researchers, including ourselves, have begun to explore alternatives to Mechanical Turk (or rely on third-party tools such as www.mTurkData.com, to continue having access to the platform). One alternative service aimed specifically at academic research is Prolific Academic (https://prolificacademic.co.uk/), which, as of January 2015, had just over 5,000 people signed up to take part in research, with just under 1,000 new recruits signing up each month.

Table 1 Popular recruitment platforms and some of their characteristics.

		Mechanical Turk	Prolific academic	
Participants	Potentially available	10k–500k	5k	
Mostly originate from	USA, India	USA, UK	
Can specify country from which to recruit?	yes	yes	
Participant reputation system	yes	yes	
Money	Fee on top of participant fee	10%–30%a	10%	
Bonus Payments possible	yes	yes (on request)	
Minimum payments	no	yes	
Access without US credentials?	no	yes	
Researcher-participant messaging	yes	yes	
Notes.

a Note that some MTurkers have a “Masters” performance-based qualification (see https://www.reddit.com/r/mturk/comments/1qmaqc/how_do_i_earn_masters_qualification/). MTurk charges researchers 30% of their participant fees for recruiting from this Masters group. Be aware that when creating a task for MTurkers to do using Amazon’s own ‘web interface’ creation tool, ‘Masters’ is set as the default group from which you wish to recruit.

Besides providing a ready source of participants, recruitment platforms also let researchers recruit from specific sub-populations. With Mechanical Turk, for example, researchers can specify whether they wish to recruit participants just from the US, or from several countries, or from anywhere in the world (permitting that there are MTurkers from those countries; one of us attempted to collect data from workers in Mexico in 2013 and failed to get participants despite posting a hit with a relatively generous payment rate). Going one step further, Prolific Academic lets researchers specify a range of criteria for recruitment, such as native language, age, sex, and even ethnicity.

Unfortunately, as alluded to before, one limitation with the existing platforms is that there is little variability in terms of the country from which one can recruit. For example, in 2010, 47% of MTurkers were North American and 34% from India (Paolacci, Chandler & Ipeirotis, 2010). It is no surprise that given the aforementioned new sign-up policy for MTurk, the percentage of US Americans taking part in recent years has become much larger (87%, Lakkaraju, 2015). Prolific Academic, on the other hand, has no such demographic blockade, with, as of 7/11/2014, participants predominantly coming from the US, UK, Ireland, and India (42%, 33%, 4%, and 2% respectively; https://prolificacademic.co.uk/about/pool; note though, that individuals without an academic email address could only sign up as from a few months ago; visit https://prolificacademic.co.uk/demographics for a live feed of demographic information pertaining to this platform’s participants). It would seem likely that the list of represented countries will grow as the platform continues to expand.

It is important to note that large swathes of potential participants from around the world still remain untapped! How would one go about recruiting participants from China, for instance, or from Colombia? Whilst sites such as TestMyBrain.org demonstrate that it is possible to recruit large numbers of participants from English-speaking countries (76%) via social networking sites and search engines (e.g., n = 4,080, in Germine et al.’s, 2012, first study), it is much harder to directly recruit only from specific countries. One option here is to create your own research panel and recruit people via the local/social media (e.g., Wilson, Gosling & Graham, 2012). A whole range of commercial software solutions exists for such a purpose; unfortunately, we are not aware of any open-source alternatives (instead, some of us have developed our own, https://github.com/ContributeToScience/participant-booking-app).

Access to a more representative sample of participants

Online research is less affected by sampling from pools of participants who can be categorized as WEIRD (Henrich, Heine & Norenzayan, 2010) than traditional lab-based research (e.g., Behrend et al., 2011; Berinsky, Huber & Lenz, 2012; Chandler, Mueller & Paolacci, 2014; Goodman, Cryder & Cheema, 2013). So what is known about the characteristics of online recruits? In terms of demographics, the ratio of female to male participants is approximately matched and the average age of the participants is currently estimated to be around 30 years of age (as found in several recent large-sampled online-studies; see Table 2 for a summary). The distribution of ages is typically ex-Gaussian (as often seen with reaction time data; Mason & Suri, 2012). Figure 2 demonstrates this right-sided long-tailed distribution from one of our own recent online studies (Woods et al., 2015). Here it is worth noting that one consequence of the much broader range of ages targeted by online research is that it makes it easier to collect data from older participants (as opposed to having to, for example, advertise in the local media for older participants as is often required in lab-based studies).

Figure 2 The distribution of ages for US and Indian participants recruited via Mechanical Turk or tested in a lab-based setting in the USA and India (Woods et al., 2015).

Table 2 Age and sex characteristics of 4 recent large internet- and phone-based sample studies.

Note that 12.5% of Mason & Suri’s (2012) participants did not report their gender.

	Recruitment platform	Sample	n	% female	Average age (SD)	
Shermer & Levitan (2014)	MTurk	US	2,737	40%	29.9 (9.6)	
Germine et al. (2012)	TextMyBrain	World	4,080 (study 1)	65%	26 (11)	
Mason & Suri (2012)	MTurk	World	2,896 (5 studies)	55%	32	
Buhrmester, Kwang & Gosling (2011)	MTurk	World	3,006	55%	32.8 (11.5)	

In terms of their ethnicity, Berinsky, Huber & Lenz (2012) contrasted US participants recruited through MTurk with a sample with a sample purported to closely match the US population at large (Matthew, Krosnick & Arthur, 2010). These researchers found that 83.5% of the Mechanical Turkers (MTurkers) were white (versus 83.0% from the general population), 4.4% were Black (versus 8.9%) and 6.7% were Hispanic (versus 5%; for interested readers, the authors also compared other differences such as marital status, income, housing state and religion and found some between-groups variations). Not restricting themselves to North Americans, Paolacci, Chandler & Ipeirotis (2010) found that 47% of MTurkers, recruited over a 3-week period in February, 2010, were from the US and 34% from India. As a side note, see K Milland (pers. comm., 2015) for a thought-provoking account on some of the hurdles faced by those outside of the US trying to earn a living via MTurk.

It is important to recognize that online participants might also have their own peculiarities. So, for example, Paolacci & Chandler (2014) have recently highlighted how, as a group, MTurkers are typically more computer literate, better educated, and less likely to be employed than the regular population; indeed, the authors argue that it is just such differences in computer literacy that may drive some of the key differences between the online participants and the population at large. Perhaps fitting with this ‘geek’ stereotype, Mason & Suri (2012) found that MTurkers tend to be less extraverted and less emotionally stable than those recruited from the street, whilst also being more open to new experiences. The authors also reported that over half of the participants whom they tested reported being on a relatively low wage (≤30,000 USD). Goodman, Cryder & Cheema (2013) directly tested how participants recruited through Mechanical Turk differed from those recruited on the street in an middle class urban area, presumably near Washington University in St. Louis (in the United States). No discernible difference was found between the groups in terms of their age, sex, or level of education. However, 27.5% of the MTurkers had English as their second language as compared to just 10.5% of those recruited from the street. The prevalence of self-reported clinical conditions such as depression matched that seen in the general population (Shapiro, Raymond & Arnell, 1997), and 95.5% of MTurkers started some form of college education (Martire & Watkins, 2015).

Thus, despite the above-mentioned variations from the general population, online participants seem to be more representative of the population at large than those typically recruited for lab-based studies, in that a broader age range, and a more equal distribution of males and females, sign up to take part in studies, who would appear to be equally susceptible to clinical conditions such as depression (which has been shown to impact on perceptual processing, Fitzgerald, 2013), but may be more educated than others in their offline community.

Here, it is also worth asking whether online participants might not be WEIRD enough to successfully complete studies? Could it be, for example, that those with too little research experience respond haphazardly, thus distorting the pattern of results that is obtained? Not so, at least not according to Chandler, Mueller & Paolacci (2014). These researchers found that over 132 experiments, 16,408 participants had undertaken an average of 2.24 studies (standard deviation of 3.19), with participants actually likely having taken part in tens or even hundreds of studies. Indeed, Rand et al. (2014) explicitly asked 291 MTurkers how many studies they had taken part in, and found that the median worker had participated in approximately 300 scientific studies (20 in the previous week; n.b. some MTurkers actively avoid academic research, R LaPlante & K Milland, pers. comm., 2015) compared to 15 studies as self-reported by 118 students as part of a participant pool in Harvard (one in the previous week).

So rather than potentially impacting results due to participant naivety, the results of research conducted online may instead be skewed because of participant overfamiliarity. Indeed, the repercussions of conducting many studies throughout the day has led to a discussion about whether certain MTurkers may not end up becoming rather ‘robotic’ in their responding (Marder, 2015). It is likely though that the field of perceptual psychology that focuses on the more automatic features of the human brain, would be less affected by such issues as compared to more cognitive fields of psychology. A recent article by the Washington Post (Searles & Ryan, 2015) nicely summarises this issue and other hot issues in the media at the moment that pertain to MTurk weirdness, and likewise conclude (amongst other things) that the source from where participants are recruited is not always critical. Another topic of concern is the high dropout rate that is sometimes exhibited by online studies (e.g., Crump, McDonnel & Gureckis, 2013). Here the interested reader is directed to Marder’s (2015) excellent article on the topic of familiarity.

It is not surprising that a variety of forums and online tools have arisen to help those taking part in online research (especially on Mechanical Turk) and one concern is that online experiments and their presumed goals are a focus of discussion via these tools (note that on some forums such as MTurkGrind, reddit.com/r/mturk, and TurkerNation, such comments are quickly reported to moderators and deleted, R LaPlante & K Milland, pers. comm., 2015). However, according to Chandler, Mueller & Paolacci (2014), whilst 28% of their 300 participants reported visiting MTurk orientated forums and blogs, it was the amount a task paid (ranked as most important) and its duration (ranked second most important) that were discussed most often as compared to, for example, a task’s purpose (which was ranked sixth).

One can wonder what the impact would be of an experimenter recruiting through their own social media channels or their lab websites, as it is likely that people similar to the experimenter are the ones likely to undertake a study so advertised (a phenomenon known as homophily, e.g., Aiello et al., 2012). Indeed, in January 2015, the Pew Research Center reported a range of substantial differences of opinion between the North American public at large and their scientific community, ranging from issues pertaining to eating genetically modified foods (88% scientists in favour, versus 37% of the general public), that humans have indeed evolved over time (98% versus 65%) and that climate change is mostly attributable to human activity (87% versus 50%). In some sense, then, one might want to consider whether we scientists might actually not be the WEIRDest of them all? Unfortunately we are not aware of any literature exploring this topic, but one needs only to turn one’s attention to Henrich, Heine & Norenzayan’s (2010) discussion of WEIRD participants for related evidence as to the impact of such issues.

In summary, although online participants most certainly have their own peculiarities as compared with the (typically American) population at large, it is doubtful whether this WEIRDness is any more pronounced than that shown by the undergraduates who take part in our lab-based research. The very fact that classical studies have been successfully replicated in both groups, each with their own peculiarities, is actually reassuring in itself.

Speed of data collection

As online research is typically conducted in parallel, a large number of participants can be recruited and take part in a study in a short space of time. With MTurk, for example, 100s of participants can sign up to take part within just 15 min of publicly releasing a study, as shown from one of our recent studies (see Fig. 3; R Pechey, A Attwood, M Munafò, NE Scott-Samuel, A Woods & TM Marteau, pers. comm., 2015). The obvious benefit is the ability to rapidly explore a given scientific issue, as demonstrated by the ‘what colour is that dress’ viral, and Tarja Peromaa’s admirable effort in collecting data from 884 participants within a few days of ‘that dress’ going public (Peromaa, 2015). As pointed out by a reviewer, do bear in mind that significant effort may still be required to link up with participant-providing services before data collection can commence. For example, it may only take 10 min to set oneself up with a Mechanical Turk account if one uses software that can automatically post experiments to this platform (see Table 3). If such software is not used, however, it can take several days to familiarise oneself with the platform in order to conduct studies manually. It may also take several weeks to create a study with an unfamiliar research software package for the first time (see Table 3). And do bear in mind that it is also important that you are on-hand to promptly answer any questions from your participants whilst the study is running (typically 0–5% of our participants email us questions).

Figure 3 The rate of experiment completion over a four-hour period (n = 360; collected February, 2015, from 8 pm onward, Eastern Standard Time; R Pechey, A Attwood, M Munafò, NE Scott-Samuel, A Woods & TM Marteau, pers. comm., 2015).

The first author’s suspicion is that ‘long tail’ sign-ups typically observed in MTurk are the result of some participants signing up and then quitting a study, and the resultant ‘time-out’ delay before a new person can take the unfinished slot.

Table 3 Popular online research platforms, their main features, strengths and weaknesses, as reported by their developers.

Survey conducted through Google Forms, on 13-3-2015, which is not listed in the below table on account of being mostly questionnaire-focused and thus ‘neutral territory’ for responders.

		JsPsych	Inquisit	LimeSurvey	ScriptingRT	Qualtrics	SoPHIE	Tatool	Unipark	WebDMDX	Xperiment	
Open-source	yes	no	yes	yes	no	citewarea	yes	no	no	yes (in beta)	
Yearly Fee for one researcher (USD)		1495			?a	?b		138.12			
Publish directly to crowd sourcing sitesc	nod	with addonse	no	no	MTurk	yes	yesf	no	nog	MTurk, ProlificAc	
Questionnaire vs Perceptual Research focus (Q vs R)											
Coding required for	Software setup	yes	no	no	yes	no	no	no	no	no	yes	
Creating a study	yes	script based	no	script based	no	no	no	no	script based	script based	
Possible trial orderings	Random	yes	yes	yes	yes	yes	yes	yes	yes	yes	yes	
Counterbalanced	yes	yes	no	no	yes	yes	yes	yes	yes	yes	
Blocked	yes	yes	yes	no	yes	yes	yes	yes	yes	yes	
Reaction times measurable in*	ms	msh	msi	ms	ms	ms	ms	ms	msh	ms	
Image, sound & video stimuli*	yes	yes	yes	yes	yes	yes	yes	yes	yes	yes	
Notes.

* Suffixed rows requested by reviewer/data not provided by developers.

a Many academic institutions have licenses with Qualtrics already. Individual academic pricing was not disclosed to us and could not be found via search engines. Note also that some features (e.g., De Leeuw (2014) more advanced randomization) may require a more expensive package.

b “Free as CiteWare, Commercial Hosting Service from SoPHIE Labs (950 USD/year)”.

c Although all platforms let the researcher provide a URL where the participant can undertake a study, some crowd-sourcing sites need to communicate directly with the testing software in order to know, for example, whether the participant should be paid.

d “None directly; but it can be used to publish on any platform that allows for custom JavaScript and HTML content”.

e See http://www.millisecond.com/support/docs/v4/html/howto/interopsurveys.htm.

f “Any crowd sourcing site that allows an external link to Tatool to run an experiment (no login required)”.

g “It uses an HTML POST command so pretty much anything, depends how skilled you are. We provide a site running a general purpose script to gather data and email it to experimenters should people not be in a position to setup a site to gather the data”.

h As discussed in the text, both these packages run outside of the browser and thus likely to more reliably and more accurately measure reaction time.

i This was a topic of contention amongst our reviewers. However, as LimeSurvey is extendable with packages such as http://www.w3.org/TR/hr-time/, timing accuracy within this framework is quite on par with other browser based frameworks.

Another benefit of the rapid collection of data is that it allows the researcher to explore the impact of day-based events, such as Valentine’s Day, Christmas, or Ramadan. Indeed, as hundreds of participants can be tested in less than an hour, this opens up the opportunity of testing individuals on even finer-grained timescales. Naturally, global time differences come into their own with such rapid sign-up.

Of course, one important caveat with the rapid large-scale sign-up of participants is that if there happens to be a flaw in one’s study, the experimenter could potentially (and rightly) receive hundreds of angry emails, each of which often require individual attention (failure to do so in the past has led to some researchers being blacklisted en-masse by MTurkers; K Milland, pers. comm., 2015). Not only should the experiment work flawlessly with no ambiguity in terms of the instructions given (as there are no experimenters present to clarify the situation), the server-hardware running the study also needs to be up to scratch. From our own experience with the first author’s Xperiment research platform (http://www.xperiment.mobi), our first studies were run from a basic Amazon Cloud server (t1.micro; see https://aws.amazon.com/ec2/previous-generation/), assuming that it could meet the demands thrown at it. However, we had not expected the sheer volume of requests to the server for one particularly demanding video-streaming study, which caused the server to crash. We now run our studies from a more substantial server (m1.small), and are in the process of providing our participants with a live public messaging tool to contact the experimenter regarding any vagaries of experimental design. Needless to say, it is particularly important with online research to pilot one’s study on several systems (perhaps with your colleagues) before releasing it to the online community. Indeed, if the study is to be done on MTurk, be advised that the platform provides a ‘testing ground’ on which you can do your own experiments, and ensure that MTurk and your software are properly communicating (https://requester.mturk.com/developer/sandbox). We also suggest that the researcher gradually increases the required sample size of their early studies (perhaps testing 10 participants with a study, then 50, then …) to ensure that their equipment can deal with the demand.

Economical

In general, collecting data from participants online provides an economical means of conducting research, with, for example, the experimenter not having to cover the fees that are sometimes associated with participants travelling to and from the lab. Those who have looked at whether the payment amount influences how many people are willing to take part (i.e., how many sign-up) and/or how seriously they take the experiment (discussed later in the section entitled ‘5. Random responding’) have, perhaps somewhat surprisingly, shown little relation between reward and effort; only the rate of recruitment seems to be influenced by payment size (Paolacci & Chandler, 2014; Mason & Watts, 2010; though do see Ho et al., 2015, who show that bonuses can sometimes improve task performance). A different picture emerges however, when you ask MTurkers themselves what motivates people to take part in low wage research. In an excellent blog article, Spamgirl, aka Kristy Milland, (Milland, 2014), highlights how low wage tasks will tend to be avoided by all, except by, for example, those using ‘bots’ to help automate their answers, and those in a desperate situation—which very likely impacts on data quality (see also Silberman et al., 2015).

We would argue that there is most certainly responsibility on the side of the experimenter to ensure fair payment2 for work done, there being no minimum wage on, for example, Mechanical Turk (see the guidelines written by MTurkers themselves for conducting research on this platform, http://guidelines.wearedynamo.org/). A sensible approach to payment may be to establish what the participant would fairly earn for an hour-long study and then scale the payment according to task duration. For example, Ipeirotis (2010) reported paying participants $1 for a 12.5 min study ($4.80/h) and Berinsky, Huber & Lenz (2012) $6/h. An online discussion by MTurkers themselves suggests that 10 cents/minute as a minimum going rate (Iddemonofelru, 2014); do note that this is below the current minimum wage in the US, which is $7.25 per hour, and 10 cents/minute is especially unfair for many of those trying to earn a living on MTurk (R LaPlante, pers. comm., 2015). A fairer rate that we have decided to adopt in our own research is 15 cents/minute (as used by the third party service www.mTurkData.com). A keen eye will spot that the wages reported above seem to increase year-by-year, which may be down to the increasing proportions of North American MTurkers compared to other nationalities. Do be advised, though, that those researchers using Mechanical Turk who are seen as offering too small a financial incentive to participants for taking part in their study are often, and rightly so, the focus of negative discussion on social media. Indeed, researchers should be aware that tools have been developed that let the MTurkers evaluate the people providing their tasks (e.g., https://turkopticon.ucsd.edu/), one of the parameters being ‘fairness’ in terms of pay (the others being ‘fast’ again in terms of pay, ‘fair’ in terms of disputes, and ‘communication’ in terms of ease of reaching the scientist; see Fig. 4 for the first author’s TurkOpticon Profile).

Figure 4 An example TurkOpticon requester profile.

An example TurkOpticon requester profile (74 MTurkers having provided feedback on the requester).

Of course, money is by no means the only motivating factor for those individuals who take part in online research. Germine et al.’s (2012) successful replication of three classical studies online (discussed earlier) were based on data collected at http://www.TestMyBrain.org, where, in exchange for partaking in the study, the participants were told how they performed in comparison to the ‘average’ participant. To put into perspective how popular this kind of approach can be, between 2009 and 2011, half a million individuals took part in a study on TestMyBrain. Indeed, the rising popularity of such ‘citizen science’ projects would, we argue here, also offer an incredible opportunity for other areas of science (see also https://www.zooniverse.org/ and https://implicit.harvard.edu/implicit/).

Cross-cultural research

The ability to write one experiment and run it with many participants from different cultures is appealing (although language translation can be effortful). For example, identifying the extent to which certain percepts are culturally-mediated has been useful for understanding a range of perceptual phenomena, including colour vision (e.g., Berlin, 1991; Kay & Regier, 2003), music perception (e.g., Cross, 2001), and, from some of us, crossmodal correspondences/expectations (e.g., Levitan et al., 2014; Wan et al., 2014a; Wan et al., 2014b; Wan et al., 2015).

Focusing on one example in more detail, Eriksson & Simpson (2010) were able to explicitly test both US and Indian participants in their study on emotional reactions to playing the lottery. Their results revealed that the female participants were less willing to enter the lottery than the male participants, though the Indian participants were generally more willing to enter the lottery than the North Americans. Their results also revealed that both male and female participants who were willing to enter the lottery gave lower ratings on how bad they would feel about losing than their counterparts who were not willing to enter the lottery. These findings allowed the researchers, at least partially, to attribute the gender difference in risky behaviour to the different emotional reactions to losing. Importantly, the researchers were able to observe the same results in two samples from different countries, and also document the cross-country difference in risky behaviour.

Complementing in-lab research

As will become apparent in the sections that follow, it is unlikely that online research will subsume everything that is done in the lab anytime soon. We believe, though, that online research can certainly provide an especially helpful tool with which to complement lab-based research. For example, if research is exploratory in nature, conducting it online first may help the researcher scope out hypotheses, and prune out those alternatives that have little support. Subsequent lab based research can then be run on the most promising subset of hypotheses. For example, our own research on crossmodal associations between basic tastes and the visual characteristics of stimuli, started out ‘online’, where we explored basic associations between these elements. After having found a link between round shapes and the word sweet, we then moved into the lab to test with real sweet tastants in order to tease out the underlying mechanisms (see Velasco et al., 2015a; Velasco et al., in press; see also Velasco et al., 2014; Velasco et al., 2015b, for another example of complementary online and offline research).

As online participants are less WEIRD than those in lab-based studies, following-up a lab-based study with one conducted online may help strengthen the generalizability of one’s initial findings or, by means of a much larger sample, offer more conclusive proof of one’s findings (e.g., Knöferle et al., 2015; Woods et al., 2013).

Comparing Online with In-lab

Whilst questionnaire-based research readily lends itself to the online environment, a common belief is that reaction-time (RT) studies, or those requiring fine temporal/spatial stimulus control cannot readily be conducted online. Perhaps surprisingly then, to date, the majority of the comparative non-questionnaire based studies that have been conducted, running essentially the same study online and in the lab, have provided essentially consistent results. Indeed, this is perhaps especially surprising, given the current replication crisis sweeping through the field of psychology (Pashler & Wagenmakers, 2012); although do consider that this preference to publish findings based on low-power statistically significant effects as opposed to insignificant effects potentially may be why the majority of online findings mirror those in lab. Of course, an alternative scenario is that when faced with significant lab findings that don’t replicate online, only the lab findings eventually get published—the insignificant findings, as sadly is often the case, getting relegated to the file drawer (Simonsohn, Nelson & Simmons, 2014; Spellman, 2012; refreshingly, the journal Psychological Science now require authors to declare, amongst other things, that “all independent variables or manipulations, whether successful or failed, have been reported in the Method section(s)”, Eich, 2014, p. 4).

Explicitly testing whether traditional lab based studies would work online, Germine et al. (2012) successfully replicated five tasks that were thought to be particularly susceptible to issues such as lapses in attention by participants and satisficing (‘cheating’, see Oppenheimer, Meyvis & Davidenko, 2009). Examples of such tasks were the Cambridge Face Memory Test, where faces are shown for three seconds to be remembered later (Duchaine & Nakayama, 2006) and the Forward Digit Span task, which is concerned with the number of digits that can be recalled after being shown serially, one after the other each for one second (Wechsler, 2008). Germine et al. (2012, p. 847) concluded that ‘...web samples need not involve a trade-off between participant numbers and data quality.’

Similarly, Crump, McDonnel & Gureckis (2013) replicated eight relatively well-established lab-based tasks online, which the authors categorised as being either RT-based (such as the Erikson Flanker task, Eriksen, 1995), focused on memory (e.g., concept learning, Shepard, Hovland & Jenkins, 1961), or requiring the stimuli to be presented for only a short period of time. The only task that was not completely replicated, a masked priming task (Eimer & Schlaghecken, 1998) was in this latter category. Here, the visual leftward or rightward pointing arrows were presented for 16, 32, 48, 64, 80, 96 ms (or so it was assumed; more on this later) and the participant’s task was to indicate the direction in which the arrows pointed. In contrast to the original lab based study by Eimer and Schlaghecken, the authors did not replicate the expected effects for stimuli of durations of 16–64 ms and concluded that short duration stimuli cannot be reliably shown when conducting internet-based research.

In a similar vein, the Many Labs study (Klein et al., 2014) directly compared 13 effects across 36 samples and settings, including multiple online samples. The online samples came from universities, from Mechanical Turk, and from a different online platform (Project Implicit) that did not pay participants. Across all of these samples, very little difference in effect size was seen between online and in-lab data.

The majority of the replication attempts by Germine et al. (2012), Crump, McDonnel & Gureckis (2013) and Klein et al. (2014) were successful. It would seem that only a subset of studies, specifically those requiring short stimulus presentation, are not so well suited to online research. Indeed, as mentioned by Crump et al., as technology improves, it is likely that even this category of task may be achieved satisfactorily online. Who knows, there may come a time when lab and online research are on a par in terms of data quality—indeed, given a disagreement between such studies, one could argue that the effects from more ecologically valid scenario, the person being tested at home in online research, should be treated preferentially as they would more likely also occur in the population at large. We will turn our attention to the issue of temporal precision later on, and demonstrate that in some circumstances, such precision can actually be achieved today.

A popular argument is that, even if online research were more prone to error than traditional lab-based research, simply by increasing the number of participants in one’s study, the researcher can offset such issues. In Simcox & Fiez’s (2014) Experiment 2, 100 MTurkers took part in a successful replication of a classic Erickson Flanker task (Nieuwenhuis et al., 2006). In order to assess how many participants would need to be tested in order to achieve an effect of a similar power to that observed in lab settings, the authors systematically varied the number of participants contributing to identical analyses (10,000 random re-samples per analysis). Reassuringly, a comparable number of online participants and in-lab participants were required for the replicated effect to be observed. The authors also noted that by increasing the sample size from 12 to 21, the chance of a Type 2 error (wrongly concluding that no effect is present) dropped from 18% to 1%—in their study, this could be achieved by recruiting additional participants for a total of $6.30. Or expressed less sensationally, by collecting 75% more participants for an additional cost of 75% of the original total participant fees.

So, although tasks requiring that visual stimuli be presented for especially short durations are seemingly less suited for online research at present, in a few years, as proposed by Crump, McDonnel & Gureckis (2013), this position will likely change, thus making such research valuable to the research community. Indeed, offsetting the reduced power of such experiments online with more participants may help us bridge the gap between now and then.

Potential Concerns with Online Research

It is important to acknowledge that there are a number of potential concerns with online research. Below we try to answer some of the most common concerns that we have encountered in our own research. Many of them, it has to be said, were raised by the inquisitive, sceptical, and in certain cases downright incredulous reviewers of our research papers.

(1) Stimulus timing

Getting a stimulus to appear on screen at the exact millisecond-specific time, and for the right duration, is indeed very hard to achieve, even for lab-based software (see Garaizar et al., 2014; Bauer, 2015); with online studies, the issue mostly boils down to the fact that the browser does not know when the monitor refreshes (although see https://github.com/jodeleeuw/jsPsych/issues/75/) and so cannot synchronize stimulus presentation with a given screen refresh. A consequence is that if a visual image is set to appear/disappear between refreshes, it will only do so on the next refresh. Indeed, if a stimulus is to appear and disappear within a period of time smaller than a refresh interval, it may not appear at all, or could appear for (often much) longer than desired, and not at the right time. This is probably why Crump, McDonnel & Gureckis (2013) were unable to replicate the Flanker task for short duration stimuli.

We tested this appearance issue in a simulation where we varied the duration of visual stimulus, starting at a random time during the refresh cycle (10,000 virtual presentations per stimulus duration). Figure 5 shows the likelihood of short duration stimuli being shown at all, or being shown for the wrong duration, or starting/stopping at the wrong time (https://github.com/andytwoods/refreshSimulation; available to run/tweak online here http://jsfiddle.net/andytwoods/0f56hmaf/). As most people use LCD monitors which typically either refresh 60 (78.1% of monitors) or 59 times a second (21.9% of monitors), we know that the majority of screens refresh every 16.67 ms or 16.95 ms (Witzel et al., 2013). As shown in Fig. 5, thus, by having none of your stimuli shown for less than 16.95 ms, the stimulus should appear on screen for about the correct duration (>90% of the time). Specifying your stimulus durations as multiples of 16.95 ms will also lead to more accurately presented longer-duration stimuli. Indeed, one may wonder why the majority of research software packages do not allow experimenters to specify their stimuli in terms of refresh intervals (as only done by DMDX, to the best of our knowledge).

Figure 5 Likelihood of stimuli of different presentation durations appearing on screen.

Likelihood of stimuli of different presentation durations appearing on screen, or doing so with the wrong start time, end time, and/or duration (screen refresh of 16.67 ms).

Auditory stimuli and the variability in the hardware they are generated by pose similar problems. For example, Plant & Turner (2009) found that computer speaker systems introduced a delay before audio presentation that ranged anywhere from 3.31 ms all the way up to 37 ms (respective standard deviations of 0.02 and 1.31 ms), with the duration of the sound varying by 1–2 ms across systems. Previous work has also found that auditory information is sometimes treated differently depending on whether participants wear headphones or hear sounds through speakers (Di Luca, Machulla & Ernst, 2009; though see also Spence, 2007). One option is that the researcher may wish to include questions pertaining to the participants’ audio hardware. Needless to say, tasks that require the fine temporal control of auditory and visual stimuli, such as needed in the visual flash illusion (Shams, Kamitani & Shimojo, 2002) and the McGurk effect (McGurk & MacDonald, 1976), would perhaps be best undertaken in the lab (if relatively few such stimuli are needed though, and latency issues with speaker presentation of sound aside, combining video and audio into videos may be an effective first step in ensuring accurate timing; the interested reader is referred to a blog article describing how timecodes can help preserve timing, http://www.bbc.co.uk/blogs/legacy/bbcinternet/2011/02/frame_accurate_video_in_html5.html). Although do consider that if such an illusion/effect were reliable enough, a staircase procedure could be used to identify the delay required for auditory and visual elements to be perceptually synchronous, which could then be used to calibrate subsequent auditory-visual testing on that computer.

(2) Reaction times

A consequence of not knowing when the screen refreshes, and thus not knowing when a stimulus will appear on the participant’s screen, is that it is hard to know from when exactly RTs should be measured (visit https://cogsci.stackexchange.com/questions/9584/how-valid-are-reaction-times-collected-from-online-studies/9967#9967 for an ongoing discussion about the quality of reaction time data collected online; do note that external hardware issues aside, sub-millisecond recording is possible through a web-browser http://www.w3.org/TR/hr-time/). Another issue is that RTs unfortunately vary quite considerably depending on the brand of keyboard used in a study, which is most certainly a big issue with online research. Plant & Turner (2009) found, for example, that the mean delay between button press and reported time was between 18.30 ms to 33.73 ms for 4 different PC keyboards (standard deviations ranged between .83 ms and 3.68 ms). With Macintosh computers, Neath et al. (2011) found that keyboards added a delay between 19.69 ms and 39.56 ms (standard deviations were between 2.67 ms and 2.72 ms). In a lab setting, this is not such an issue where typically participants are tested using the same experimental apparatus and thus same keyboard (example 5 ms response delays with a random variation of −2.7 ms to +2.7 ms are 22.11, 18.07, 17.59, 20.9, 22.3 ms; mean = 20.19, stdev = 2.23). However, when lots of different keyboards are used, a whole variety of different latencies act to introduce troublesome variation (example 5 random response delays of 20 ms to 40 ms, with the same random variation added are 19.45, 37.8, 37.57, 22.7, 31.23 ms; mean = 29.75, stdev = 8.43).

All is not lost however. Systematically exploring how well RTs could actually be measured online, Reimers & Stewart (2014) recently tested RTs on 5 different computers, 3 web-browsers, and 2 types of web-technology (Adobe Flash, HTML 5) using a Black Box Toolkit (http://www.blackboxtoolkit.com/; a piece of hardware that can be used to accurately measure response times and generate button presses). The authors used the device to detect screen flashes generated by the testing software by means of a photodiode, and to generate button presses at precise times by completing the circuit of a button of a hacked keyboard. Although there was some variability across machines, and although RTs were generally overestimated by 30 ms (comparable to the delays reported above, although standard deviations were typically 10 ms), the authors concluded that the noise introduced by such technical issues would only minimally reduce the power of online studies (the authors also suggested that the within-participant design is particularly well-suited to online research given this variability). Once again bolstering the support for conducting valid RT research online, Schubert et al. (2013) found comparable RT measurement variability when comparing their own online Flash-based research software ScriptingRT (mean 92.80 ms, standard deviation 4.21) with lab-based software using the photodiode technique mentioned above (millisecond mean reaction times for DMDX, E-prime, Inquisit and Superlab, were respectively 68.24, 70.96, 70.05, 98.18; standard deviations 3.18, 3.30, 3.20 and 4.17; the authors must be commended for their ‘citizen science’ low-cost Arduino-based timing solution, which Thomas Schubert fleshes out on his blog https://reactiontimes.wordpress.com/). These findings were broadly mirrored by De Leeuw & Motz (in press) who compared accuracy for recording RTs in a visual search task that run either via Matlab’s Psychophysics ToolBox or in a webbrowser via JavaScript. Whilst RTs for the latter were about 25 ms longer than the former, reassuringly there were no real differences in data variability over platforms. Simcox & Fiez (2014) found that browser timing accuracy was only compromised when unusually large amounts of system resources were in use. The authors measured timing by externally measuring screen flashes with a photodiode that were placed 1000 ms apart in time, and concluded that browser based timing is in most scenarios as accurate as lab-based software. In summary, then, it would seem then that the variability introduced by participants using different computers/monitors/web-browsers, is negligible in comparison to the variability introduced by the participants themselves (Brand & Bradley, 2012; Reimers & Stewart, 2014), although, one has to wonder whether using the participant’s smartphone-camera to detect screen refresh/stimulus presentation parameters (from say a few pixels devoted to this purpose in the top of the participants’ screen) and appropriately feeding this knowledge back to the testing software may help with accuracy. Some modern-day cameras certainly are able to capture video at high enough frame rates (e.g., 120 Hz, Haston, 2014).

One way to get around the browser-related limitations of not knowing when the screen refreshes is to ask participants to download experimental software to run outside of the browser (unfortunately MTurk does not permit the downloading of external software; Prolific Academic does allow this if sufficient explanation is provided for the participants). One problem here though is that the experimenter cannot really ask their participants to undertake the fine calibrations normally required to set up experimental lab-based software (e.g., timeDX, http://psy1.psych.arizona.edu/~jforster/dmdx/help/timedxhtimedxhelp.htm), so more superficial means of calibration must be automatically undertaken. Seeing if their own compromise solution was sufficient for the downloadable webDMDX, Witzel et al. (2013) tested whether the results of classical time critical studies differed across lab based DMDX (Forster & Forster, 2003) and webDMDX and found good consistency across software platforms. Curiously, however, the results of a stimulus that had been set up to appear for 50 ms in the lab-based software matched those for a 67 ms duration stimulus in the web based software. The authors found that the lab-based stimulus was just over 3 refreshes in length (16.67 ms * 3 = 50.01 ms) and so was actually shown for an additional interval, for 66.68 ms, as was 67 ms stimulus (n.b., DMDX rounds to the nearest refresh interval), which was easily corrected. Thus, if your participants trust your software, and your participant panel permits it, it may be advisable to use software like webDMDX for those experiments requiring fine temporal control of stimuli.

(3) Variability in hardware

Perhaps the most obvious issue with online research, as alluded to above, is the sheer variety of hardware and software used by participants. Although it can be argued that online research is more ecologically valid because of this varied hardware compared to lab-based studies that all run on the same device, hardware variability, nevertheless, poses some unique challenges for the experimenter; especially when considering that the web browser can only determine a few device parameters such as screen resolution and operating system (but see Lin et al., 2012). For example, the resolutions of monitors differ massively over participants; we found in 2013 an average resolution of 1,422 × 867 pixels across 100 participants’ monitors, with large respective standard deviations of 243 and 136 pixels (Woods et al., 2013). As there is no way to assess the physical size of monitors via a web browser, standardising the size of one’s stimuli over participants is extremely difficult. As a workaround, C Bechlivanidis & DA Lagnado (pers. comm., 2015) had their participants hold up a CD, a credit card, or a 1 US dollar bill to their screen, and then adjust a shape on the screen to match the size of the object (see also Yung et al., 2015). The authors also asked their participants whether they were an arm’s distance away from their monitor to get an idea of their distance from the monitor (see also Krantz, Reips & Bosnjak (2001), who suggests a real world ‘rule of thumb’—by holding your thumb an arm’s distance from the monitor, perpendicular elements directly beneath the thumb are approximately 1 or 2 visual degrees). Another approach is to find your participant’s blind spot—by asking them to focus on a shape whilst another shape horizontally moves relative to it, and indicate when the moving shape disappears from view—and then resize experimental images appropriately. Sadly though, we cannot anchor our online participants’ heads in place to prevent fidgeting, although, as suggested by a helpful audience member in a recent talk by the first author, monitoring the participant via a webcam and resizing stimuli appropriately may be one future strategy to help cope with this.

Another issue is that the many dials and buttons that adorn the modern-day computer often make it impossible to quantify properties such as volume, brightness, and colour. There are, though, ways to counter this issue. For example, the participant could be asked to adjust the volume until an audio stimulus is just audible, or indicate when elements in a visual image have the highest contrast (To et al., 2013). Yung et al. (2015) did the latter by presenting a band of grey bars on screen and having their participants adjust the brightness of the bar (in their software) until all of the bars were visible. We have also started to include an audio password (or AudibleCaptcha) in our experiments that can only be answered when the volume is set appropriately (Knöferle et al., 2015). The daring may even consider using staircases to establish a variety of thresholds for audio stimuli. Although it is impossible to really control for background noise levels, by using webcam microphones, it may be possible to quantify background noise levels and re-run noisy trials or add noise levels as a covariate in subsequent data analyses.

Perhaps one of the greatest challenges is colour. One approach to combating this issue is to use colour words instead of the colours themselves (e.g., Piqueras-Fiszman, Velasco & Spence, 2012; Velasco et al., 2014); though, of course, this solution is only going to be suitable for a small number of studies (those that only use colour categories). An initially promising solution would be to run studies on identical devices such as the same generation iPad device. Unfortunately, however, even purportedly identical screens viewed under identical environmental conditions vary in terms of colour and brightness (Cambridge Research Systems, S Elliott & R Ripamonti, pers. comm., 2015). Others have suggested using psychophysics to identify issues with the current monitor and then dynamically adjusting the presented images appropriately. Our hats come off to To et al. (2013); they presented a variety of coloured and hashed line patches in different shades and had their participants adjust their properties so that, for example, two such patches would match in terms of their brightness. The authors found that participants performed to a similar ability to a photometer (.5% sensitivity difference). A potential future solution could be to ask participants to use the camera on their mobile devices to video both their computer screen being used for a study, and a common, colourful, household object, (e.g., a bottle of CocaCola™; cf. the size solution of C Bechlivanidis & DA Lagnado, pers. comm., 2015). Software on the mobile device could then potentially liaise with the research software to calibrate screen colour to the reference object. Thus, although presenting the same colour to participants irrespective of device is probably not achievable with current technologies, we believe that there are some nice ‘work-arounds’ that may help somewhat offset any variability in one’s data due to inconsistent colour (as can also be done by collecting data from many more participants).

Summarising briefly, the variability in hardware used by participants in online studies pose unique problems that with the current level of technology are hard to directly address. Several workarounds exist for each issue however, and at the end of the day, collecting more data (as always) is a healthy way to offset some of these issues. Of course, hardware-related data variability can be factored out in subsequent data analyses if a within-participants design can be used over a design where groups of participants’ data (and differing variation due to hardware variability) are contrasted against each other.

(4) Unique participants?

How can you be sure that the same participant is not taking part in the experiment multiple times? Participants recruited through Mechanical Turk or Prolific Academic must have an online profile that at least theoretically prevents them from taking part in the same study more than once. Although potentially an individual can have multiple accounts, it is harder to do these days with increasingly tight security-conscious sign-up criteria. Indeed, if the participant wishes to get paid, they must provide unique bank account and Social Security Number details (for MTurk), each of which requires a plethora of further identification checks (as does PayPal, which Prolific Academic currently uses for participant payment).

The research software itself can also provide some checks for uniqueness, for example, by storing a unique ID in each participant’s web browser cache or Flash cache, thus making it easier to identify repeat participants. Although it is sometimes possible to identify potential repeaters by means of their (identical) IP address, Berinsky, Huber & Lenz (2012) noted that the 7 out of 551 participants in their Mechanical Turk study who had identical IP addresses could well have performed the study on the same computer, or same shared internet connection; indeed, this day and age, the participants could even have done the study through the same Virtual Private Network and be in quite different geographic locations from those determined via IP address (or indeed through self-report).

A related concern arises when an experimenter conducts multiple different online experiments using the same platform. Preventing previous participants from participating in future experiments is difficult using MTurk (but see http://mechanicalturk.typepad.com/blog/2014/07/new-qualification-comparators-add-greater-flexibility-to-qualifications-.html), so typically the experimenter ends up having to manually, tediously, exclude repeats after participation. Bear in mind here that relying on participants to not undertake a task if they have done a similar one in the past is unfair given the sheer number of past studies each likely will have undertaken. Perhaps a much more impactful issue is when participants become overly familiar with popular experimental methods/questionnaires that are used by different researchers. Highlighting this issue, Chandler, Mueller & Paolacci (2014) found that out of 16,409 participants in over 132 studies, there were only 7,498 unique workers with the most active 1% completing 11% of hits (see also Berinsky, Huber & Lenz, 2012; N Stewart, C Ungemach, AJL Harris, DM Bartels, BR Newell, pers. comm., 2015).

Although these issues most certainly are a concern for researchers focusing on the study of perception, it is likely that repeat participants would be far more problematic for more cognitive-focused areas of psychology. It may simply be the case for the psychologist interested in perception to ask participants how often they have undertaken similar tests in the past and use this data as a covariate in their subsequent statistical analysis.

(5) Random responding

A common concern with online research is that those taking part in a paid study do not do so with the same care and diligence as those in a lab-based study. In fact, however, the research that has been conducted in this area to date shows that lab-based studies are not necessarily the gold standard that is often presumed. In one such study, conducted by Oppenheimer, Meyvis & Davidenko (2009), immediately after completing two classic judgement and decision-making studies, participants were presented with a catch-trial in which they were explicitly told to click a small circle at the bottom of the screen, as opposed to 1 of 9 response buttons making up a line scale that was shown in the centre of the screen. Not only did a disquieting 46% of the participants fail the task, but only by excluding these individuals were both the classic tasks successfully replicated (interestingly, in a second study, individuals had to redo the task until they got it correct—performance on the subsequent classic task no longer varied as a function of being ‘caught out’ in the catch task). Thus one cannot necessarily assume that the participants in lab-based research are attending as carefully as one might hope. As an example from our own experiences, one author received a text message from such a participant who was ‘mid study’, saying he would be late for his later experiment! Reassuringly though, perhaps again highlighting that perceptual psychology is more robust to such issues than other areas of our discipline, some of us used the Oppenheimer, Meyvis & Davidenko (2009) attention check for an online face emotion task and found that only 1% of MTurkers failed the task (Dalili, 2015; see Hauser & Schwarz (in press) for an in depth discussion). We return to this issue below.

Perhaps one key issue scientists have with online research is the absence of the experimenter who can be quizzed to clear up uncertainties, or to make sure that the participant follows the instructions. Painting a bleak picture, Chandler, Mueller & Paolacci (2014) asked 300 MTurkers what they were doing whilst completing a study, and found that 18% of responders were watching TV, 14% listening to music and 6% were communicating with others online (the interested reader is directed to a video where an MTurker discusses this issue in reference to looking after her baby whilst participating in research, Marder, 2015). Several strategies, besides the catch trial mentioned earlier (Oppenheimer, Meyvis & Davidenko, 2009), have been developed to deal with the consequences of such distraction and potential disinterest (Downs et al., 2010; Crump, McDonnel & Gureckis, 2013; Germine et al., 2012), perhaps the simplest being to quiz the participants as to the nature of the task before proceeding to the study. Crump, McDonnel & Gureckis (2013) found that this approach led to a close replication of a classic rule-based classification learning study (Nosofsky et al., 1994), compared to an earlier study where there was no such intervention (as would also seem to be demonstrated in Experiment 2 of Oppenheimer, Meyvis & Davidenko, 2009, mentioned in the preceding paragraph).

Indicating that this issue of distraction is not such a problem, when Hauser & Schwarz (in press) directly set about comparing the performance of lab-based and internet-recruited participants on Oppenheimer, Meyvis & Davidenko’s (2009) catch trial, found the latter group much less likely to fail at the task. Hauser and Schwarz first found that lab-based participants failed an astounding 61% of the time—even more than the original study—whilst online participants recruited on MTurk only failed 5% of the time. This broad pattern of results was replicated for a novel version of the catch trial in Experiment 2. To test whether MTurkers were just very vigilant for such catch trials (as they may have had similar ones in the past; see the ‘overfamiliarity’ discussion above) or whether, indeed, MTurkers paid more attention, in a third study, both groups were tested on a soda-pricing task (once again adapted from Oppenheimer, Meyvis & Davidenko, 2009) that has been shown to be sensitive to levels of attention. Supporting the latter account, the online participants scored much better in a test that was sensitive to attention levels than their lab-based counterparts.

In summary, whilst the lack of experimenter supervision for participants recruited online most certainly is worrying, it is important to bear in mind that lab-based research does not necessarily ensure attentive participants either. The very fact that a lot of past research has been replicated would indicate that the different issues with online and in lab research may be similarly impactful on results in our field.

(6) Ethics

While it is relatively clear where the responsibility lies for ethics in a study conducted within a given department, online research is often an unknown area for both the researcher and the local ethics committee. The British Psychology Society have weighed in on this topic (British Psychological Society, 2006; British Psychological Society, 2013; see also the American Psychological Association’s overview on this, Kraut et al., 2004; Ross, 2014), highlighting the key issue that it is the physical absence of the experimenter during the study, preventing, for example, the experimenter from stopping the study early if the participant starts showing any signs of distress. Presumably though, the online participant would feel less obligation to actually finish a study that they were uncomfortable with, compared to if it were a lab-based study.

There are several other issues as well (for a timely special issue focused on ‘ethical issues in online research’, see James & Busher, 2015). Besides issues of fair pay (highlighted earlier), online anonymity is also a key issue. For example, with a bit of deduction, it is often possible to extrapolate the identity of an individual from their pattern of responses (El Emam & Arbuckle, 2013; King, 2011; see also some such high-profile examples from the Netflix challenge, Narayanan & Shmatikov, 2008, and social networks, Narayanan & Shmatikov, 2009). Highlighting this, MTurker Worker IDs are made available to research software when people take part in an MTurk study. We asked 100 MTurkers to enter their Worker ID into Google and tell us “Did your Google search of your Worker ID find links to ‘raw data’ (answers you gave) from previous studies you have taken part in?” and “Did your Google search results contain information that revealed your name or IP address?” A staggering 47 MTurkers reported finding such past data in their search results, whilst 5 MTurkers reported finding their name/IP-address. Further exploration is warranted to check just what information past researchers are making publicly available online alongside potentially identity revealing MTurker Worker IDs, as this clearly goes against ethical guidelines. Several MTurkers also emailed to say that their past Amazon store reviews of books appeared in their search results—with a bit of investigation it transpired that Amazon Ids and MTurker Worker IDs are one and the same (see Lease et al., 2013, who discuss this and other issues in detail)! In light of the above, we would urge researchers to carefully select the information that is stored alongside collected data, and to remove Worker IDs before sharing data online. If Worker ID data must be stored online (e.g., to be shared by the members of a specific lab), that data should be adequately encrypted, and not left as ‘plain text’ as was seen often in the just mentioned survey.

The recent drive to open-source datasets coupled with the ethical requirement to allow participants to withdraw their data after data collection (up to a certain point in time, anyway, such as at the conclusion of the analysis) unfortunately muddies the waters regarding anonymity. One strategy for this we have adopted is to ask participants to provide a password that can be used if they wish their data removed by a later date; although given the large number of passwords one must remember these days, it is not clear if this will prove effective. The School of Experimental Psychology in Bristol University has provided a carefully thought-out example of a Participant Information Sheet and Consent Sheet package which takes this and other issues into account (https://osf.io/buj5i/). Given the ever changing landscape of online research we feel (along with an anonymous reviewer) that the BPS and APA perhaps need to re-assess their policies in light of the above anonymity revelation, and on a regular basis, following in the footsteps of Bristol’s well-applauded example ethics documentation.

Software to Conduct Research Online

In 2004, the lead author devoted several months creating a one-off website to run a JavaScript-powered study on crossmodal face perception using videos (J Stapleton, AT Woods, S Casey, FN Newell, pers. comm., 2015; see Thompson, 1996, for an early pre-JavaScript experiment run entirely in HTML). Things have progressed a great deal since then! There are now a variety of software platforms aimed at collecting questionnaire-based data online, with a smaller number of packages now aimed specifically at conducting online behavioural research. Some of the latter, alongside their strengths and weakness as reported by their main developers, have been listed in Table 3.

One way in which the packages differ is in terms of whether they are open-source or commercial in nature. Whilst licensed software is often thought to be easier to use and have a better support network than open-source software, a downside is that if a bug is found, you are at the mercy of a limited number of developers to fix the problem, instead of being able to immediately explore the code yourself or to ask for help from the lively open-source community. Conventional wisdom would also suggest that commercial software would be easy and more versatile than open-source ‘freely contributed to’ software but the reality is that this is often not the case (e.g., The Gimp, http://www.gimp.org/, is an open-source feature rich alternative to Adobe Photoshop). Moreover, commercial software is typically configured with the needs of large-scale corporate users in mind, whereas the open-source community may be more receptive to suggestions that benefit academic users.

If you have no programming experience, deciding on a testing package that does not require the coding may be a quicker option for getting a study on the web, if your task only requires typical experimental features (such as buttons, scales, the ability to show pictures, etc.). Some packages such as Qualtrics and Tatool, for example, let you create studies by dragging and dropping components on screen. An intermediate option offering more flexibility is to use software that relies on scripts to run an experiment (e.g., ScriptingRT, WebDMDX, Xperiment).

Whether or not the research software is Adobe Flash based or not is another consideration. Although Flash has purportedly been ‘dying’ for a number of years now (not so, according to @AS3lang), it is, in fact, still present on most modern computers; for example, it is installed automatically within the Chrome web browser which has 61.9% market share (W3Schools, 2015), and can be automatically installed in other popular browsers such as Firefox (23.4% market share). Flash is also making a comeback by re-inventing itself as a cross-platform tool capable of making apps for both Android and IOS; indeed, it won the 2015 Consumer Electronics Show best mobile application development platform. As we found out recently, though, given the lead author’s package called Xperiment, reliance on the proprietary closed-source Adobe Flash environment meant that when bugs in closed source code did arise, we were entirely dependent upon Adobe engineers to fix the issues. At the start of 2014, Adobe updated their software and thus ‘broke’ an experiment we were running, leading to a loss of 31.3% of participant data (see the bug here https://productforums.google.com/forum/m/#!topic/chrome/ifL98iTMhPs). This may well have been due to ‘teething issues’ due to Google Chrome releasing its own version of Flash around that time called ‘pepperFlash’. In light of this, though, the lead author is porting over the Xperiment package to the open-source cross-platform Haxe toolkit, which allows software to natively run in the browser (without Flash), as well on several platforms such as IOS and Android.

On the Future of Online Perception Research

We believe that smart devices will come into their own in the coming years (e.g., Brown et al., 2014; Dufau et al., 2011; Miller, 2012). Making them seem particularly well suited for online perceptual psychology are their plethora of sensors (light levels, global position system, proximity) and actuators (vibration, flashing light), as well as their range of peripherals such as smart watches (letting you measure for example, heart rate, cadence, and even sleep quality), other wearables such as motion tracking (e.g., http://www.xensr.com/) and even intelligent cushions that measure seated posture quality (http://darma.co/). Of course, these new technologies may well be affected by the same issues that were highlighted before. For example, in our own tentative steps down the road of smart phone research, we have found large differences in terms of the levels of vibration that different smartphones can produce, which is, presumably attributable to the devices using a variety of vibration motors.

Not only are smart devices rich in sensors and actuators, they can add a new dimension to research by being able to contact participants at any point during the day using Push notifications, or to link with software services to provide even richer sources of information for your investigation. If a study were concerned say, with vibration detection in noisy environments, the device could be made to vibrate only when the background noise level was at a desired level. Alternatively, GPS could be used if your paradigm required participants only be tested in certain geographical locations. We predict such ‘mashups’ of technologies (e.g., Paredes-Valverde et al., 2015) will really be a game changer for perceptual psychology (for some predictions on future ways we will interact with our devices, see Nijholt, 2014).

Unfortunately, the current state of affairs mirrors that for online research in 2005 where one-off experiment apps must be made, typically for either IOS or Android devices. An early example of such an app, reported in 2010 by Killingsworth & Gilbert, asked users randomly throughout the day a series of questions about their current levels of mind wandering and happiness. Curiously, the authors found that mind-wandering was negatively associated with happiness (although more recent findings suggests that this effect depends upon the mind wandering being negative itself in terms of emotion, Poerio, Totterdell & Miles, 2013).

Conducting research on gaming devices such as the Xbox One and Playstation 4 is surprisingly not that far away. Transpilers, or source-to-source compilers (Wikipedia, 2015) allow developers to write code once and port that code to different platforms and programming languages. A transpiler that can currently port code to such gaming devices is the commercially available Unity 3D package (http://unity3d.com/; see also Adobe Air and the open-source Haxe platform; as of yet though, neither package can port to gaming devices; https://www.adobe.com/uk/products/air.html, http://haxe.org/; however, this is coming soon to Haxe as discussed in talks by Robert Konrad and Lars Doucet at the World Wide Haxe conference in Paris, May, 2015, http://wwx.silexlabs.org/2015/).

‘Big data’ is most certainly part of the future for psychological research, where hundreds of thousands of participants contribute data as opposite to tens or hundreds as seen in typical lab-based studies. To attract these numbers, researchers, for example, gamify their paradigm to make it fun for people to take part, offer feedback about how people have done after task completion (e.g., testMyBrain that we mentioned earlier providing score feedback). An alternative strategy is to piggyback existing sources of data, as Stafford & Dewar (2014) nicely demonstrate with their n = 854,064 study exploring skill learning whilst people played an online game called Axon, that was developed for the Wellcome Trust (http://axon.wellcomeapps.com/).

In China, around 10:34 pm on Thursday 19, 2015 (The Chinese New Year), apparently 810 million smartphones were shaken at their TVs every minute. Over the course of a 4-hour long show (China Central Television Spring Festival gala), the shake count totalled an incredible 11 billion! What had happened was that weChat (a Chinese instant messaging service) in collaboration with a plethora of retail companies had offered the public the possibility of winning ‘red envelopes’ containing small amounts of money (for example, 2 Chinese Yuan, or about 0.32 USD), by just shaking their phones. One can only wonder what could be achieved if an intrepid researcher managed somehow to build upon this, a la Stafford & Dewar (2014). Care would be needed to ensure such a study was ethically sound, however (c.f. the recent Facebook emotion manipulation study, as discussed by Ross, 2014).

Conclusions

Over the last 5 years or so, we have found the internet to be a very effective means of conducting online research to address a number of perceptual research questions. It offers a number of potential benefits over in-lab testing and is particularly useful for quickly collecting a large amount of data across a relatively wide range of participants. On the flip-side, there are a number of potential limitations that also need to be borne in mind. In terms of ethics, it seems that online participants are not as anonymous as they should be, which needs addressing. It is also tricky to account for differences in hardware over machines and there still remain some issues related to the fine control of timing. Over the coming years though, it is likely that such issues will become less of a problem as technology develops, new solutions arise, and clearer ethical guidelines become available to researchers. In the meantime, a simple approach to deal with some of these issues though is, as always, to collect more data, which fortunately is easy, economical and fast in online research. As with the introduction of the personal computer to the psychology lab, we feel that online research will revolutionize the way in which we investigate perceptual processes.

Supplemental Information

Data S1 Data pertaining to 100 Mechanical Turkers and their anonymity

Click here for additional data file.

We are grateful for feedback on an earlier draft of this manuscript by Rochelle LaPlante, Kristy Milland, Marcus Munafò, and Six Silberman.

Additional Information and Declarations

Competing Interests

Author Contributions

1 Although the reasons for this are still unknown, a first wave cull in 2012 had been thought to be due issues of fraud regarding unscrupulous MTurkers gaming the system (Admin, 2013). It is the first author’s belief, though, that the 2014 change occurred as pressure had been put on MTurk to ensure all its workers were tax identifiable.

2 Tangentially, the lead author of this paper was approached after a recent talk given on the topic of this manuscript and was queried as to whether it was fair to pay online participants more than others do in their online studies, as this would presumably drive up the cost for all. Although a valid point, it is the lead author’s view that such a payment ethos is not fair on the participants themselves.

Andy Woods is the founder and employee of Xperiment, Surrey, United Kingdom.

Andy T. Woods conceived and designed the experiments, performed the experiments, analyzed the data, wrote the paper, prepared figures and/or tables, reviewed drafts of the paper.

Carlos Velasco, Carmel A. Levitan, Xiaoang Wan and Charles Spence wrote the paper, reviewed drafts of the paper.

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
