# Peer review of "Conducting perception research over the internet: a tutorial review"

_PeerJ, doi:10.7717/peerj.1058_

## Round 0.1 · original submission · Major Revisions

Please carefully consider reviewer comments and use discretion whether applicable to make revisions. I struggle with aspects of focus here which could be due to the number of authors each contributing her/his share with tighter control needed over content synthesis. I have a few observations to add to the reviewers.

It seems to be called a tutorial and then a lit review and then heavy enough emphasis on MTurk to indicate that should be in the title. It doesn't address enough historical aspects leading up to current trends.

I created the first Web-based survey and study of Internet addiction back in 1995 for online participants who perceived themselves as addicted and I published it in 1996. While it is pretty much lost today, for what it is worth, perhaps anecdotally, you can still access it at http://w8r.com/iads/survey.html. Times and methods have certainly changed; the importance of your study is certainly timely and relevant.

In addition to reviewer suggestions, I would like to see either more comparison over time regarding the matter of online surveys for psychology experiments in lab v. online or clearer indication that your study seeks to look at a more definitive timeline, should I have missed it.

While I would posit the overall decision between Minor-Major here, I wish to ensure all relevant reviewer comments are seriously taken into consideration in revision by all authors for the best synthesized paper to be published in PeerJ.

All the best.

·

Basic reporting

Very clearly reported with some exceptions as described in the general comments.

Experimental design

The research question is defined but should be more focused as described in the general comments.

Validity of the findings

The conclusions are balanced and the authors offer many useful suggestions and tipps for conducting online studies.

Additional comments

In the article “Conducting perception research over the internet: a tutorial review” authors give the authors provide a review of methodological sutides on online testing.

While the question of is important I believe that the authors need to focus more strongly on the topic of perception research. In the current form the review reiterates many discussions without adding new aspects that are irrelevant to perception-studies.

# Major issues

In my view the whole section on "Benefits of conducting research online" mixes very general issues and those specifically relevant to perception-research. I am not convinced that the same aspects are relevant to perception studies compared with questionnaire studies. Rather than repeating these well-known studies on benefits of online testing - particularly mTurk - this should be severely shortened (1 page) or removed altogether.

On the balance the parts that are most relevant for anyone planning a study - sections on timing-issues, hardware and selection on software could be expanded.

# Specific remarks:

# Benefits of conducting research online

## Access to a more representative sample of participants

- The authors make a claim that the online sample fit better, but offer only very vague data about the sample. It would make a much better point if they scrutinized a random selection of papers reporting on lab-based studies. Otherwise authors should remove claims about lab-based research.
- Authors should try ot focus on aspects that are specific to perception studies.
- It is unclear what the sample should be representative for. Only US-citizens?

## Access to large pools of participants

- "highly praised on twitter, " should be removed.
- This paragraph actually describes how internet based recruitment may be more objective than lab-based forms.


## Recruitment platforms

- This section would also be much better if a stringent criteria were used to comapre the different services. Availibility from outside US, diversity of sample, sample-size, princing


## Speed of data collection

- A more balanced view would try to describe the time it takes to set up an account for the first time etc. and start an individual study.

## Economical

## Cross-cultural research

- "(e.g., Levitan et al., 2014; Wan et al, 2014a, 2014b, 2015)." It is unclear why the authors choose to citate studies that all pertain to modalities that cannot be tested online
- "Eriksson and Simpson": Why was this study chosen? It seems like a questionnaire study.

## Complimenting in-lab research

- please correct "complImenting"

## Comparing online with in-lab

- "indeed, given a disagreement between such studies": This is a very bold statement because problems replicating e.g. backward masking are very likely due to technical problems presenting stimuli online rather than an artifact that is induced by lab-equipment.
- Here, too a more structured summary of the three comparison studies may be beneficial.

# Potential concerns with online research

## 1 Timing

- This section may be divided in presentation and response timing.

## 2 Variability in hardware

- Again this could be divided in presentation and response-timing.
- Also it may be best to use different modalities to structure the paragraph.
- Why cannot the McGurk effect presented as a video?
- It should be noted how relevant such problems are for within-sj designs

## 3 Unique participants?

## 4 Random responding

- "Oppenheimer, Meyvis, and Davidenko": As the authors note these authors study decision making. Is this also a pertinent problem for perception studies?

## 5 Ethics

- Authors should note solutions to the dilemmata described, especcially those raised by larger Societies.

# Conducting research online

- Consider to change the title of the subsection to software.
- The comparison of tools is great and could be much expanded. It would be important to know whether RTs can be collected, or not?

- "Questionnaire vs Research focus": What is meant by this?

Reviewer 2 ·

Basic reporting

The submission is well written and structured.

Experimental design

Not really applicable to this paper, as it is a review of experimental design.

Validity of the findings

This is a very useful and timely review that will benefit the field.

Additional comments

P. 5: Do smartphones count as "computers"?
P. 7: Where is the lab-based data from? Are the "samples" equivalent for the comparisons shown? Certainly some labs study ageing, such as changes across the lifespan! Also, do the WEIRD aspects of online samples cause more concern than noted here? Certainly there is potential for overcoming WEIRD samples, but it is unclear that is yet the case.
P. 9: On homophily: I understand the example you note, but is there any evidence this is the case for online studie? That seems most relevant.
P. 11: On Recruitment Platforms: I wonder if this section should come far earlier. The sections that precede it discussion Mechanical Turk in a familiar tone, yet only here is it really explained in detail. Explaining it first would be helpful perhaps.
P. 28: Interesting study described here -- but did they also debrief the 46% that missed the extra task to see if they really missed it, or if instead they just ignored it thinking it was a trick to see if they would not focus on the primary task?
P. 30-31: Do the societies (BPS, APA, etc) need to revisit these guidelines (or lack thereof)? One criticism is raised, but it seems there other things missing from those policies as well. The anonymity issue, clashing with the requirement to allow later data deletion, is not just an online problem of course, and a suggestion would be interesting here.

---

## Round 0.2 · Minor Revisions

Please respond to the reviewer's latest round of comments, which are few, prior to a final decision on this manuscript.

Thank you.

·

Basic reporting

-

Experimental design

-

Validity of the findings

-

Additional comments

The authors have done a good job in addressing most of my previous issues. While I would still prefer a tighter focus, I do not think this is an issue that calls for a rejection. Since the target audience is now more clearly described, readers will be able to decide for themself.

I have two specific remarks on additions to the manuscript that should be addressed in a revision.

# Specific remarks:

First, authors now note that "significant effort" is necessary before one can collect data. In order not to put manypotential user off, it would be important to provide some estimate on how long it takes to learn/teach these techniques. Even if there is no hard data on this some information based on your experiences would be good. For example a friend of mine regularly teaches workshops for online-questionnaire studies that take approx. 4h and include learning lime-survey and specifics of online questionnaire research.
Could you provide a similar estimate for the time it takes to set up an Mturk account and how much time is beeing spend on such a study, e.g. how many percent of the participants try to get in contact?

Second, in table 3 that gives an overview of the software it is claimed that all allow for a measurement of reaction times. I think this is misleading because Limesurvey and Inquisit are clearly not on par on this issue. It may be much better to describe the precision at which reaction times can be measured (for limesurvey seconds; for Inquisit several milliseconds). This will help researchers decide on the adequate tool.

---

## Round 0.3 · accepted · Accept

Thank you for the valuable time and patience you put into making your manuscript pass the peer review process successfully. I am pleased to inform you that your manuscript is accepted into PeerJ with expectation as noted below.